# Neurotrophins in Zebrafish Taste Buds

**DOI:** 10.3390/ani12131613

**Published:** 2022-06-23

**Authors:** Claudia Gatta, Valentina Schiano, Chiara Attanasio, Carla Lucini, Antonio Palladino

**Affiliations:** 1Department Veterinary Medicine and Animal Production, University of Naples Federico II, Via F. Delpino 1, 80137 Naples, Italy; gattaclaudia@gmail.com (C.G.); valentinaschiano8742@gmail.com (V.S.); chiara.attanasio@unina.it (C.A.); 2Department Agricultural Sciences, University of Naples Federico II, Via Università 100, 80055 Portici, Italy; antonio.palladino@unina.it

**Keywords:** nerve growth factor (NGF), brain derived neurotrophic factor (BDNF), Neurotrophin 3 (NT3), Neurotrophin 4 (NT4), taste buds (TBs)

## Abstract

**Simple Summary:**

Zebrafish is a powerful vertebrate model organism, whose similarities with mammals are fundamental to validate its use for experimental purposes. In this study, the authors demonstrate the presence of neurotrophic factors, namely neurotrophins, in numerous taste bud cells of this fish. The reported results suggest an essential role of these factors in taste bud function. Interestingly, the results described in this study are in accordance with those reported in some mammalian species. Therefore, despite the different anatomical characteristics of the anterior digestive tract in mammals and fish, the taste buds maintain similarities in both shape and functional mechanisms in the two classes.

**Abstract:**

The neurotrophin family is composed of nerve growth factor (NGF), brain-derived neurotrophic factor (BDNF), Neurotrophin 3 (NT3) and NT4. These neurotrophins regulate several crucial functions through the activation of two types of transmembrane receptors, namely p75, which binds all neurotrophins with a similar affinity, and tyrosine kinase (Trk) receptors. Neurotrophins, besides their well-known pivotal role in the development and maintenance of the nervous system, also display the ability to regulate the development of taste buds in mammals. Therefore, the aim of this study is to investigate if NGF, BDNF, NT3 and NT4 are also present in the taste buds of zebrafish (*Danio rerio*), a powerful vertebrate model organism. Morphological analyses carried out on adult zebrafish showed the presence of neurotrophins in taste bud cells of the oropharyngeal cavity, also suggesting that BDNF positive cells are the prevalent cell population in the posterior part of the oropharyngeal region. In conclusion, by suggesting that all tested neurotrophins are present in zebrafish sensory cells, our results lead to the assumption that taste bud cells in this fish species contain the same homologous neurotrophins reported in mammals, further confirming the high impact of the zebrafish model in translational research.

## 1. Introduction

Neurotrophins comprise nerve growth factor (NGF), brain-derived neurotrophic factor (BDNF), Neurotrophin (NT) 3 and NT4 [1] acting through two classes of receptors: p75 and Tyrosin kinase (Trk). P75 is a pan-neurotrophin receptor that regulates both cell survival and death, depending on the cellular context and its interaction with other receptors; however, its functions are still far from being fully elucidated [2]. There are three Trk receptors: TrkA for NGF, TrkB for BDNF and NT4, and TrkC for NT3 signals. In axon terminals, the endocytosis of neurotrophins bound to Trk receptors forms signalling endosomes acting locally. Additionally, being retrogradely transported to remote cell bodies, they promote neuron survival, morphogenesis, and maturation [3]. In addition, neurotrophins are active in regulating the development and maintenance of taste buds in mammals [4,5,6]. Indeed, mice with gene deletion for BDNF, NT4, NT3 and TrkB displayed the reduction of gustatory papillae and taste buds [7,8,9].

In fish, neurotrophins comprise NGF, BDNF, NT3 and NT4 homologous to those of mammals. A further neurotrophin, NT6, is exclusively reported in fish. This latter NT probably derives from the duplication of an ancestral gene also giving rise to two paralogs, NT6 and NGF [10]. In fish, neurotrophins are widely represented in the brain [11,12,13,14,15,16,17,18,19] and sensory organs such as the retina [20,21,22,23,24], inner ear, lateral line and olfactory organ [25,26,27]. Previous studies in fish demonstrated the presence of neurotrophin receptors in taste buds (TBs), namely the common Trk amino acid sequence in sea bass [28], while TrkB and TrkA have been reported in zebrafish (*Danio rerio*) [27,29,30], thus suggesting the sensibility of fish TBs to neurotrophins.

TBs in fish are numerous and, contrary to mammals, have both an internal and external location. External TBs, widespread in the skin of the head and sometimes of the body, as in the case of catfish lips and barbels, detect food in proximity. Oropharyngeal TBs, instead, seem to play the most important role in the final choice of food, as suggested by a usual fish-eating behavior featured by the rejection of food after its ingestion [31]. Despite the different localization, fish TBs are morphologically similar to mammal ones, being pear- or onion-shaped. Specifically, in the oropharyngeal cavity of zebrafish, TBs protrude from the epithelium (Figure 1A), and, according to the electron density, they contain elongate dark cells with many small microvilli and light cells with a single large microvillus. Underlying this structure there are basal cells, which resemble Merkel cells and are horizontally located (Figure 1B). In zebrafish, another type of cell similar to dark cells, featured by several small microvilli that form a brush-like apical ending, was reported by Hansen and collaborators [32]. At the interface between taste bud cells and epithelial cells, there are small semilunar-shaped cells named marginal cells.

Therefore, the aim of this study is to investigate if NGF, BDNF, NT3 and NT4 are also present in the taste buds of zebrafish, further enhancing the translational impact of this model organism.

## 2. Materials and Methods

For the study, adult females and males (9 months) (*n* = 4) of *Danio rerio* were employed. Wild-type AB zebrafish were housed in ZebTEC semi-closed recirculation housing systems (Techniplast, Buguggiate, VA, Italy), and water conditions were kept constant at a temperature of 28 °C, pH (7.5) and conductivity (500 μS) on a 14/10 light/dark cycle. The experimental protocols were conducted according to the Italian Decree 26/2014 and were approved by the Institutional committee of the University of Naples Federico II (Centro Nazionale delle Ricerche, n°2/2020-PR). Fish were anaesthetized by 0.1% ethyl 3-aminobenzoate, methane sulfonate (Sigma Chemicals Co., St. Louis, MO, USA), and heads were collected.

Histological and immunohistochemical techniques were previously described [33]. Briefly, the sections were dewaxed and incubated with 0.3% hydrogen peroxide for 30 min at room temperature (RT) to block endogenous peroxidase activity. Then, the sections were rinsed in 0.01 M phosphate-buffered saline (PBS), pH 7.4, for 15 min and subsequently incubated for 20 min at RT with normal goat serum. Then, rabbit polyclonal antibodies against NGF (H-20, sc-548 Santa Cruz Biotechnology, Inc., Santa Cruz, CA, USA), BDNF (N-20, sc-546 Santa Cruz Biotechnology, Inc., Santa Cruz, CA, USA), NT-3 (N-20, sc-547, Santa Cruz Biotechnologies, Santa Cruz, CA, USA) and NT-4 (N-20, sc-545, Santa Cruz Biotechnology, Inc., Santa Cruz, CA, USA) were diluted at 1:150; 1:200; 1:200 and 1:200, respectively (13). After incubation with primary antisera, the sections were rinsed in PBS for 15 min and incubated for 30 min at RT with EnVision+System-HRP, Labelled Polymer anti rabbit (Dako, Santa Cruz, CA, USA). Subsequently, the sections were rinsed in PBS for 15 min and then incubated for 30 min at RT with avidin-peroxidase complex. Peroxidase activity was detected using a solution of 3-3′ diaminobenzidine tetrahydrocloride (Sigma, St. Louis, MO, USA) of 10 mg in 15 mL 0.5 M Tris buffer, pH 7.6, containing 0.03% hydrogen peroxide.

To study the distribution of cells that were immunoreactive to neurotrophins, we analyzed eleven levels of the oropharyngeal cavity. Each level is separated from the previous one by an approximate distance of 200 µm. By this approach, five consecutive sections are placed on a different slide, each stained by hematoxylin and eosin (HE) and immunocytochemically against NGF, BDNF, NT3 and NT4. To obtain the anterior-posterior distribution of neurotrophin cell populations, we counted all immunoreactive cells to each neurotrophin in all stained slides. Furthermore, only taste buds that were recognizable in each seriated slide were counted to determine the percentage of different neurotrophin positive cells. The results are omitted in cases of absence of positive cells for the specific antibody.

All the stained sections were photographed using a Leica DMRA2 microscope. The digital raw images were optimized for contrast and illumination by using Adobe Photoshop CS5 (Adobe Systems, San Jose, CA, USA).

## 3. Results

All the four antisera employed showed immunoreactivity in the cells of taste buds localized along the entire oropharyngeal cavity, situated both in the roof and floor epithelium. NGF and BDNF positive cells were generally thin cells, often separated from each other by other negative cells. They could be morphologically ascribed to dark and light cells (Figure 2A–E).

NT3 and NT4 positive cells were thin elongated cells or round cells located in the basal region of the taste bud, often grouped strictly together. They could be ascribed to dark, light and Merkel-like cells. In addition, NT3 immunoreactivity was detected in cells situated at the taste bud periphery; therefore, they could be considered marginal cells (Figure 3A–F).

Considering the pattern of distribution of NGF, BDNF, NT3 and NT4 in taste bud cells along the oropharyngeal cavity, it is clear that their number increases in more posterior zones, particularly in the case of BDNF positive cells (Figure 4).

## 4. Discussion

In our study, all antisera against the investigated neurotrophins (NGF, BDNF, NT3 and NT4) showed immunoreactive cells in taste buds of the oropharyngeal cavity of zebrafish. The distribution of TB positive cells was not uniform along the oropharyngeal cavity, increasing toward the posterior region, especially in the case of BDNF positive cells. Within the taste buds, the morphology of positive cells suggests that all the detected neurotrophins are localized in sensory (light and dark) cells. NT3 and NT4 were also observed in Merkel-like cells, meaning that these cells may hold a mechanoreceptive or neuroendocrine potential. In addition, NT3 seems to be present in marginal cells, which fall on the boundary between taste buds and epithelium [32].

In our previous studies, the same antisera, analyzed by western blotting, showed bands whose molecular weights were ascribable to both precursors and/or mature forms of neurotrophins [13,34,35], depending on the tissue considered. In this study, we did not isolate taste buds for obvious technical limitations; therefore, we are not able to define whether the immunoreactivity is ascribable to neurotrophin precursors and/or mature forms.

A previous study on zebrafish showed the presence of TrkA and TrkB/BDNF in sensory cells of taste buds [29,30]. In view of this, a local action in zebrafish taste buds could be supposed for NGF as well as for BDNF/NT4 detected in the present study for TrkA and TrkB positive cells, respectively, probably through a paracrine/autocrine mechanism. Similarly, in mammalian taste buds, NGF [34], BDNF [36] and NT4 [6] were identified along with their specific receptors TrkA [34] and TrkB [6].

Furthermore, Germanà and collaborators [26] reported that in zebrafish, TrkC receptors are only localized in fibers innervating TBs, leading to the hypothesis that NT3 in TBs may act on their specific innervation. Similar findings were reported in hamsters [37], contrary to what has been displayed in mice [6], in which the TrkC receptor was not detected in either taste buds or nerves, while NT3 resulted in being largely represented in taste buds. To explain the findings observed in mice, the authors hypothesized that NT3 may act with a low affinity to TrkA and TrkB receptors, indicating a mechanism suitable for being potentially also applied to zebrafish.

## 5. Conclusions

In conclusion, despite the different environments in which zebrafish and mammals live, as well as the different anatomies of the anterior digestive tract and distributions of TBs, in TBs zebrafish display the same homologous neurotrophins reported in mammals. The fast cell turnover that features TB cell populations and regulates their function is probably regulated in a paracrine/autocrine manner. This statement further confirms the reliability and significance of zebrafish as a model organism.

## Figures and Tables

**Figure 1 animals-12-01613-f001:**
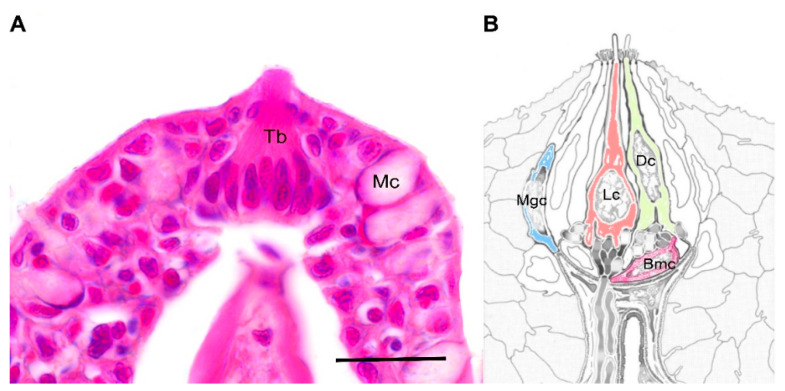
Taste bud. (**A**) Histological section of a single taste bud protruding from the pluristratified epithelium of the oropharyngeal cavity. Hematoxilin-eosin staining; (**B**) Schematic representation of the ultrastructural organization of fish taste buds (modified from Hansen et al. 2002). Abbreviations: Tb = taste bud; Mc = mucous cell; Dc = dark cell (green); Lc = light cell (red); Bmc = basal cell (pink); Mgc = marginal cell (blue). Scale bar = 20 µm.

**Figure 2 animals-12-01613-f002:**
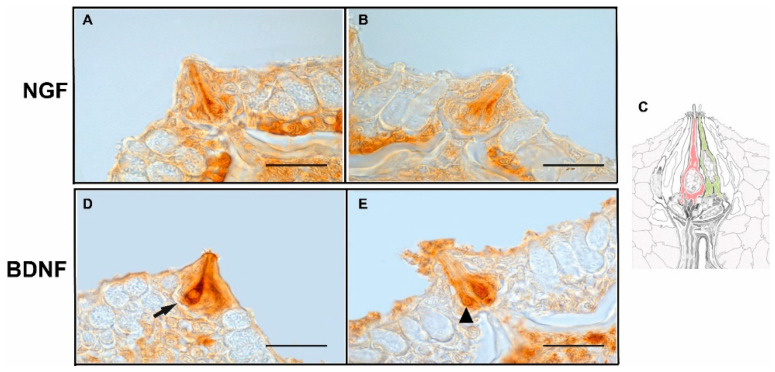
(**A**,**B**) NGF and (**D**,**E**) BDNF positive cells. (**C**) Taste bud scheme showing hypothetic congruence of positive cells with ultrastructural classification. The arrow indicates a putative light cell, while the arrowhead indicates a dark cell. Refer to color code in Figure 1 to recognize the cell type. Scale bar = 20 µm.

**Figure 3 animals-12-01613-f003:**
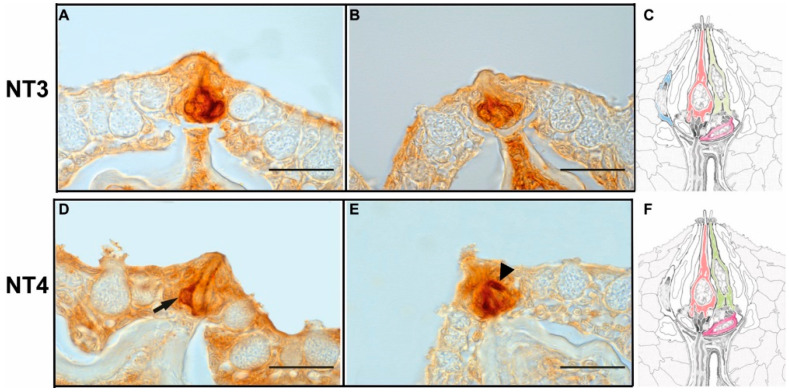
(**A**,**B**) NT3 and (**D**,**E**) NT4 positive cells. (**C**,**F**) Taste bud scheme showing hypothetic congruence of positive cells with ultrastructural classification. The arrow indicates a putative light cell, while the arrowhead indicates a dark cell. Refer to color code in Figure 1 to recognize the cell type. Scale bar = 20 µm.

**Figure 4 animals-12-01613-f004:**
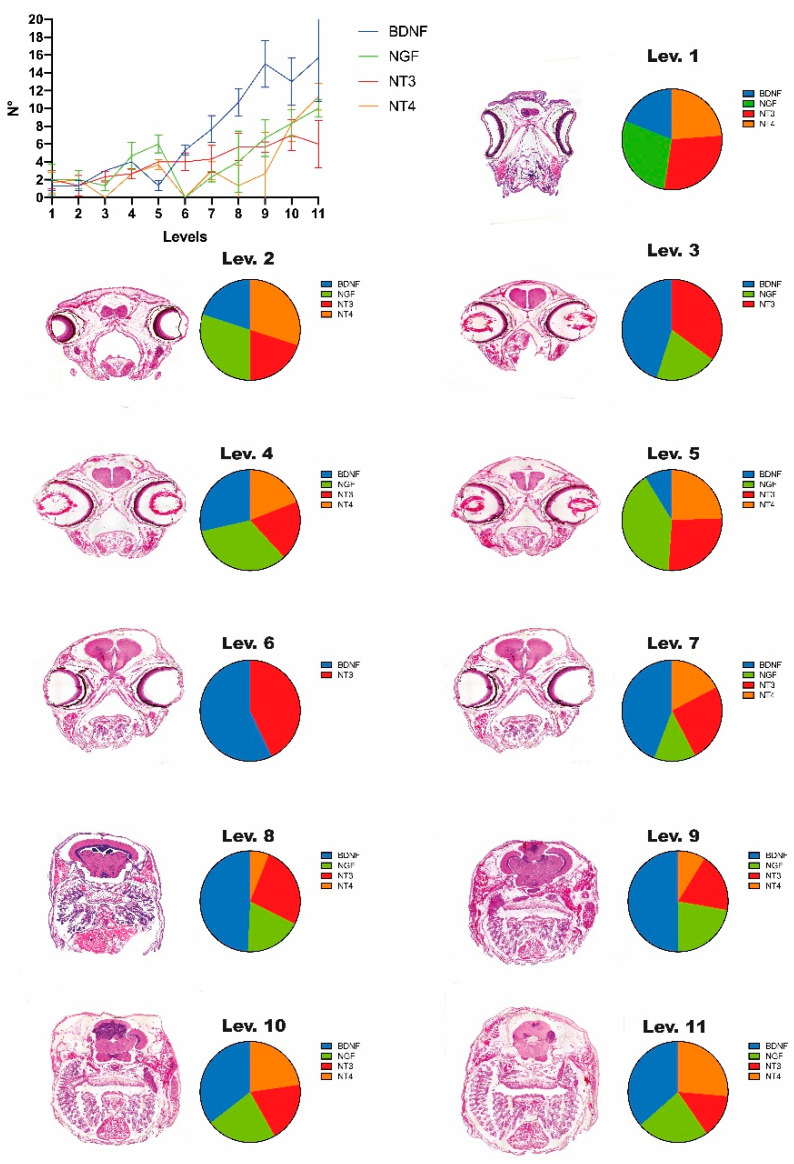
Distribution of NGF, BDNF, NT3 and NT4 positive cells in the oropharyngeal cavity. Diagram showing the number of positive cells for each transversal section of oropharynx. Level 1–Level 11: Diagrams showing the percentage of neurotrophin cells in the taste buds totally comprising the serial sections that compose the level. On the left of each diagram, the transversal sections of the zebrafish head (EE stain) representative of the level are reported (for details, see Materials and Methods section).

## Data Availability

The data presented in this study are available in “Neurotrophins in zebrafish taste buds”.

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
