# Peer review of "Neurotrophins in Zebrafish Taste Buds"

_animals, 2022, doi:10.3390/ani12131613_

Round 1

Reviewer 1 Report

The authors characterized expression of neurotrophins including NGF, BDNF, NT3 and NT4 in the taste buds of zebrafish through histological and immunohistochemical techniques. They conclude that all the tested neurotrophins are present in the zebrafish sensory cells and taste bud cells in this fish species contain the same homologous neurotrophins reported in mammals.

I don’t think this paper is acceptable for publication in this journal. 1) The significance of the study is not clear. 2) The conclusion is not supported by the data. 3) The Materials and Methods are too simplified. 4) The manuscript is badly written. There are too many writing errors and long sentences in the manuscript.

Line 27, suggesting

Line 36, two class of

Lines 40-41, “Regarding Trk receptors, TrkA transduces NGF signals, TrkB 40 BDNF and NT4 ones while TrkC from NT3.” This sentence should be revised.

Lines 41-44, Long sentence, should be revised.

Line 42, bounded

Lines 48-51, Long sentence, should be revised.

Line 54, aminoacidic

Lines 57-60, Long sentence, should be revised.

Author Response

The Authors thank the referee for the comments that led to improve the manuscript. Please, find point by point response

The authors characterized expression of neurotrophins including NGF, BDNF, NT3 and NT4 in the taste buds of zebrafish through histological and immunohistochemical techniques. They conclude that all the tested neurotrophins are present in the zebrafish sensory cells and taste bud cells in this fish species contain the same homologous neurotrophins reported in mammals.

I don’t think this paper is acceptable for publication in this journal.

  • The significance of the study is not clear.

The manuscript is a morphological study of taste buds in zebrafish. Because this fish is an important model, every scientific statement regarding its characteristics could be a useful tool for future investigations. 

2) The conclusion is not supported by the data.

The text of the conclusion has been changed

3) The Materials and Methods are too simplified.

The description of materials and methods have been extended

4) The manuscript is badly written. There are too many writing errors and long sentences in the manuscript.

The text was carefully checked and too long sentences were shortened  

Line 27, suggesting

Line 36, two class of

Lines 40-41, “Regarding Trk receptors, TrkA transduces NGF signals, TrkB 40 BDNF and NT4 ones while TrkC from NT3.” This sentence should be revised.

Lines 41-44, Long sentence, should be revised.

Line 42, bounded

Lines 48-51, Long sentence, should be revised.

Line 54, aminoacidic

Lines 57-60, Long sentence, should be revised.

All minor requested changes were done

Reviewer 2 Report

 Based on immunohistochemical techniques, the authors attempted to examine if mammalian neurotrophin homologs  exist in the taste bud of zebrafish to evidence the value of zebrafish in translational research. The overall quality of this submission can be considered for publication if the authors explain how they constructed the distribution pattern of four types of neurotrophins shown in figure 4. As one tissue section was supposed to be incubated with four primary polyclonal antibodies and tissue positions of all neurotrophins were visualized all in similar red color using the same peroxidase-conjugated second antibody, how the distribution of each type of neurotrophin in one section was determined? Can the authors reconstruct the distribution pattern and clearly illustrate the localization of every type of neurotrophin in different color? The position of taste bud was not found in figure 1 B.

Author Response

The Authors thank the referee for the comments that led to improving the manuscript. Please, find point by point response

Based on immunohistochemical techniques, the authors attempted to examine if mammalian neurotrophin homologs  exist in the taste bud of zebrafish to evidence the value of zebrafish in translational research. The overall quality of this submission can be considered for publication if the authors explain how they constructed the distribution pattern of four types of neurotrophins shown in figure 4. As one tissue section was supposed to be incubated with four primary polyclonal antibodies and tissue positions of all neurotrophins were visualized all in similar red color using the same peroxidase-conjugated second antibody, how the distribution of each type of neurotrophin in one section was determined? Can the authors reconstruct the distribution pattern and clearly illustrate the localization of every type of neurotrophin in different color? The position of taste bud was not found in figure 1 B.

The authors hope that the paragraph added in the Material and methods section could exhaustively answer the reviewer’s questions: “To study the distribution of cells immunoreactive to neurotrophins, we analyzed eleven levels of the oropharyngeal cavity. Each level is separated from the previous one by an approximative distance of 200 µm. By this approach, five consecutive sections, are placed on a different slide each stained by hematoxylin and eosin (EE) and immunocytochemically against NGF, BDNF, NT3, and NT4. To obtain the anterior-posterior distribution of neurotrophin cell populations, we counted all immunoreactive cells to each neurotrophin in all stained slides. Furthermore, only taste buds that were recognizable in each seriated slide were counted to determine the percentage of different neurotrophin positive cells. Results are omitted in case of absence of positive cells for the specific antibody”.

Reviewer 3 Report

The manuscript entitled "The highly conserved pivotal role of neurotrophins in taste buds: evidence in zebrafish” has the purpose, declared by the Authors, of investigating the localization of NGF, BDNF, NT3 and NT4 in the taste buds of zebrafish.

The presentation of the manuscript needs to be improved to be considered for publication.

-The title is suitable for a review more than an article, in my opinion.

-Why do the Authors decide to investigate only the localization of NGF, BDNF, NT3 and NT4 and not also their receptors?

-Because of their choice, moreover, they should remove from the Keywords the number 5 Tyrosine kinase (Trk) receptor.

-Line 51-52 Add Review: de Girolamo P, D'Angelo L. Neurotrophins in the Brain of Teleost Fish: The State of the Art. Adv Exp Med Biol. 2021;1331:289-307. doi:10.1007/978-3-030-74046-7_20.

Materials and Methods

-Line 67 How many specimens have been analyzed? Specify the number of fish.

-Describe in detail the housing conditions of the fish. It is known in fact, that the average life span as well as the time spent inside the taste buds is highly temperature dependent, for example.

-Lines 90-96: the sentence from line 90 to line 96 should not be included in the results section.

In the results section, Authors they should stick to the writing the results of their research.

-Line 104: remove “generally” from the text.

-Line 105 “NGF and BDNF positive cells were “slender”. What’s the significance of that, I wonder.

The Figures are of good quality. However, some aspects need to be improved:

-Figure 2: in A-B, D-E indicate light and dark cells with arrows or arrow heads.

-If NGF and BDNF are both located in light and dark cells, remove C or F. Two identical schematic representations are useless.

-Figure 3: in A-B, D-E indicate with arrows and arrow heads the different cell types.

-Lines 122-127: how did the Authors proceed to establish the NGF, BDNF, NT3 and NT4 distribution pattern in taste bud cells along the oropharyngeal cavity? Based on which criteria do they state that the number of immunoreactive cells is increasing in more posterior zones, particularly in the case of BDNF positive cells?

-What are the landmarks chosen in the observation of the 11 sections? Obviously, we need to know where to take observations, namely the so-called "landmarks” or the specific anatomical points, to make sure you have observed the taste buds of the same region in all specimens.

-Describe in detail what criteria were used to identify each observation field.

-How many sections of the same sample were observed?

-Was the tridimensionality of the structure analyzed considered?

-The bidimensional vision observed under the microscope makes it necessary to observe the entire taste buds on histological serial sections. Because of the pear- or onion-shape of the taste buds a section does not include all the cell types that make it up. The selected section plane can exclude a cell that may appear in later sections, jumping to hasty conclusions, with falsified values.

-Lines 124-127: the expressed concept is redundant; it has already been said (lines 122-124).

-The Authors declare that BDNF positive cells are the prevalent cell population in the posterior part of oropharyngeal region. How do the authors explain this result?

-Much of the discussion has focused on receptors which have not been analyzed in this paper, so it seems inadequate for the context of the manuscript itself. Rewrite.

-The authors could further update the manuscript literature. I found in fast research this paper that should be evaluated by authors for inclusion in the manuscript:

Aragona M, Porcino C, Guerrera MC, Montalbano G, Laurà R, Levanti M, Abbate F, Cobo T, Capitelli G, Calapai F, Vega JA, Germanà A. Localization of BDNF and Calretinin in Olfactory Epithelium and Taste Buds of Zebrafish (Danio rerio). Int J Mol Sci. 2022 Apr 23;23(9):4696. doi: 10.3390/ijms23094696.

Germanà A, Sánchez-Ramos C, Guerrera MC, Calavia MG, Navarro M, Zichichi R, García-Suárez O, Pérez-Piñera P, Vega JA. Expression and cell localization of brain-derived neurotrophic factor and TrkB during zebrafish retinal development. J Anat. 2010 Sep;217(3):214-22. doi: 10.1111/j.1469-7580.2010.01268.x.

Line 144 delete “;”

It would be very interesting for the readers if the Authors discussed their hypothesis of turnover of taste buds cells.

Author Response

The authors thank the referee for the comments that led to improving the manuscript. Please, find point by point response

The manuscript entitled "The highly conserved pivotal role of neurotrophins in taste buds: evidence in zebrafish” has the purpose, declared by the Authors, of investigating the localization of NGF, BDNF, NT3 and NT4 in the taste buds of zebrafish.

The presentation of the manuscript needs to be improved to be considered for publication.

-The title is suitable for a review more than an article, in my opinion.

The title was changed to “Neurotrophins in zebrafish taste buds”

-Why do the Authors decide to investigate only the localization of NGF, BDNF, NT3 and NT4 and not also their receptors?

 We didn’t look at receptor localization since it has already been reported by Germanà e al. Neuroscience Letters 354 (2004) 189–192. The study was cited in the Introduction and reported in References.

-Because of their choice, moreover, they should remove from the Keywords the number 5 Tyrosine kinase (Trk) receptor.

 According to this observation, the keyword was removed.

-Line 51-52 Add Review: de Girolamo P, D'Angelo L. Neurotrophins in the Brain of Teleost Fish: The State of the Art. Adv Exp Med Biol. 2021;1331:289-307. doi:10.1007/978-3-030-74046-7_20.

The reference and citation in the text have been added

Materials and Methods

-Line 67 How many specimens have been analyzed? Specify the number of fish.

We added the number of fish (n =3) according to this comment.

-Describe in detail the housing conditions of the fish. It is known in fact, that the average life span as well as the time spent inside the taste buds is highly temperature dependent, for example.

Description of housing conditions has been added

-Lines 90-96: the sentence from line 90 to line 96 should not be included in the results section.

We modified the text accordingly by moving this sentence to the introduction section.

In the results section, Authors they should stick to the writing the results of their research.

The results start with the phrase “All the four antisera….”

-Line 104: remove “generally” from the text.

Done

-Line 105 “NGF and BDNF positive cells were What’s the significance of that, I wonder.

In line with the comment “Slender” was removed

The Figures are of good quality. However, some aspects need to be improved:

-Figure 2: in A-B, D-E indicate light and dark cells with arrows or arrow heads.

According to this suggestion, we added the indications to Figures 2 and 3.

-If NGF and BDNF are both located in light and dark cells, remove C or F. Two identical schematic representations are useless.

To overcome this redundancy, in line with the previous comment, Figure 2 was remodulated

-Figure 3: in A-B, D-E indicate with arrows and arrow heads the different cell types.

As stated before, in order to improve overall image clarity we added arrows and arrowheads.

-Lines 122-127: how did the Authors proceed to establish the NGF, BDNF, NT3 and NT4 distribution pattern in taste bud cells along the oropharyngeal cavity? Based on which criteria do they state that the number of immunoreactive cells is increasing in more posterior zones, particularly in the case of BDNF positive cells?

-What are the landmarks chosen in the observation of the 11 sections? Obviously, we need to know where to take observations, namely the so-called "landmarks” or the specific anatomical points, to make sure you have observed the taste buds of the same region in all specimens.

-Describe in detail what criteria were used to identify each observation field.

-How many sections of the same sample were observed?

-Was the tridimensionality of the structure analyzed considered?

-The bidimensional vision observed under the microscope makes it necessary to observe the entire taste buds on histological serial sections. Because of the pear- or onion-shape of the taste buds a section does not include all the cell types that make it up. The selected section plane can exclude a cell that may appear in later sections, jumping to hasty conclusions, with falsified values.

The authors hope that the paragraph added in the Material and methods section could exhaustively answer the reviewer’s questions: “To study the distribution of cells immunoreactive to neurotrophins, we analyzed eleven levels of the oropharyngeal cavity. Each level is separated from the previous one by an approximative distance of 200 µm. By this approach, five consecutive sections, are placed on a different slide each stained by hematoxylin and eosin (EE) and immunocytochemically against NGF, BDNF, NT3, and NT4. To obtain the anterior-posterior distribution of neurotrophin cell populations, we counted all immunoreactive cells to each neurotrophin in all stained slides. Furthermore, only taste buds that were recognizable in each seriated slide were counted to determine the percentage of different neurotrophin positive cells. Results are omitted in case of absence of positive cells for the specific antibody”.

-Lines 124-127: the expressed concept is redundant; it has already been said (lines 122-124).

We removed the sentence reported in lines 124-127 accordingly.

-The Authors declare that BDNF positive cells are the prevalent cell population in the posterior part of oropharyngeal region. How do the authors explain this result?

Please, refer to the previous response

-Much of the discussion has focused on receptors which have not been analyzed in this paper, so it seems inadequate for the context of the manuscript itself. Rewrite.

The sentences concerning the receptors were shortened

-The authors could further update the manuscript literature. I found in fast research this paper that should be evaluated by authors for inclusion in the manuscript:

Aragona M, Porcino C, Guerrera MC, Montalbano G, Laurà R, Levanti M, Abbate F, Cobo T, Capitelli G, Calapai F, Vega JA, Germanà A. Localization of BDNF and Calretinin in Olfactory Epithelium and Taste Buds of Zebrafish (Danio rerio). Int J Mol Sci. 2022 Apr 23;23(9):4696. Doi: 10.3390/ijms23094696.

Germanà A, Sánchez-Ramos C, Guerrera MC, Calavia MG, Navarro M, Zichichi R, García-Suárez O, Pérez-Piñera P, Vega JA. Expression and cell localization of brain-derived neurotrophic factor and TrkB during zebrafish retinal development. J Anat. 2010 Sep;217(3):214-22. Doi: 10.1111/j.1469-7580.2010.01268.x.

The references and citations in the text have been added

Line 144 delete “;”

We thank the Reviewer for having identified this careless mistake, we deleted “;”.  

It would be very interesting for the readers if the Authors discussed their hypothesis of turnover of taste buds cells.

In our study, we did not mention the turnover of taste bud cells. However, it is a remarkable point worth future studies devoted to.

Round 2

Reviewer 1 Report

The manuscript has been considerably improved.

Author Response

The authors thank the reviewer for the kind revisioning of the manuscript

Reviewer 2 Report

Animals-1696317 has been carefully revised and the reviewer's question on figure 4 regarding the analysis of different types of neurotropins in zebrafish taste buds  based on immunostaining has been answered in the revised materials and methods. The revised manuscript now can be accepted for publication. 

Author Response

(The authors gave the same response as above.)

Reviewer 3 Report

The manuscript has been improved compared to the previous version.

The manuscript is clear and presented in a well-structured manner, now.

The cited references are most recent publications and relevant.

The title was changed but there is a gap between the title reported in the responses to the Reviewer (Neurotrophins in zebrafish taste buds) and that reported in the manuscript (Neurotrophin expression in zebrafish taste buds). It is recommended to choose between one of the two following titles: “Neurotrophins in zebrafish taste buds” or “Neurotrophins localization in zebrafish taste buds”.

The section "Materials and methods" has been completed, although there is a discrepancy between the number of samples reported in the Reviewer responses (n. 3) and what is included in the text (n. 4). How many samples have been analyzed?

The figures/schemes are appropriate. They show properly the data. They are easy to interpret.

The conclusions are consistent with the evidence and arguments presented.

Author Response

The authors thank the reviewer for the careful revision. We have changed the title of the manuscript to "Neurotrophins in zebrafish taste buds". Moreover, we used 4 animals, as is specified in the text. We apologize for the mistakes.